# Scalable solution for delivery of diabetes self-management education in Thailand (DSME-T): a cluster randomised trial study protocol

Chaisiri Angkurawaranon [ORCID],[1] Iliatha Papachristou Nadal,[2] Poppy Alice Carson Mallinson,[2] Kanokporn Pinyopornpanish,[1] Orawan Quansri,[3] Kittipan Rerkasem,[4,5] Supattra Srivanichakorn,[6] Win Techakehakij,[7] Nutchanath Wichit,[8] Chanapat Pateekhum,[1] Ahmar H Hashmi,[1] Kara Hanson,[9] Kamlesh Khunti,[10] Sanjay Kinra[2]

For numbered affiliations see end of article.

**Correspondence to**
Dr Chaisiri Angkurawaranon;
chaisiri.a@cmu.ac.th

## ABSTRACT

**Introduction** Type 2 diabetes mellitus is among the foremost health challenges facing policy makers in Thailand as its prevalence has more than tripled over the last two decades, accounting for considerable death, disability and healthcare expenditure. Diabetes self-management education (DSME) programmes show promise in improving diabetes outcomes, but this is not routinely used in Thailand. This study aims to test a culturally tailored DSME model in Thailand, using a three-arm cluster randomised controlled trial comparing a nurse-led model, a peer-assisted model and standard care. We will test which model is effective and cost effective to improve cardiovascular risk and control of blood glucose among people with diabetes.

**Methods and analysis** 21 primary care units in northern Thailand will be randomised to one of three interventions, enrolling a total of 693 patients. The primary care units will be randomised (1:1:1) to participate in a culturally-tailored DSME intervention for 12 months. The three-arm trial design will compare effectiveness of nurse-led, peer-assisted (Thai village health volunteers) and standard care. The primary trial outcomes are changes in haemoglobin A1c and cardiovascular risk score. A process evaluation and cost effectiveness evaluation will be conducted to produce policy relevant guidance for the Thai Ministry of Public Health. The planned trial period will start in January 2020 and finish October 2021.

**Ethics and dissemination** Ethical approval has been obtained from Thailand and the UK. We will share our study data with other researchers, advertising via our publications and web presence. In particular, we are committed to sharing our findings and data with academic audiences in Thailand and other low-income and middle-income countries.

**Trial registration number** NCT03938233.

> ### Strengths and limitations of this study
>
> ► A three-arm cluster randomised controlled trial to evaluate clinical and cost-effectiveness of a culturally tailored diabetes self-management education (DSME) under two alternative modes of delivery (nurse-led and peer-assisted) will provide policy makers with options for scalability.
> ► A culturally tailored DSME programme has been developed with input from stakeholders (policy makers, clinicians, nurses, village health volunteers and people with diabetes).
> ► A series of short films have been developed to introduce key topics, as there is increasing recognition that films are a highly efficient medium for communicating information, particularly in low literacy settings.
> ► The study will be conducted in two provinces of Thailand, so some caution may be required when generalising findings to the rest of the country.
> ► Due to the nature of the intervention, blinding of participants to their trial arm will not be possible.

prevalence has more than tripled over the last two decades to an estimated 4 million adults (age adjusted prevalence 7.1%) living with diabetes in 2015.[1,2] Diabetes is associated with several macrovascular (eg, ischaemic heart disease) and microvascular complications (eg, nephropathy, retinopathy, neuropathy and foot disease), which primarily account for the considerable death and disability (of which diabetes is the fifth leading cause in Thailand).[3] In addition, diabetes in Thailand causes a twofold increase in healthcare expenditure and significant loss of economic productivity—of both people with diabetes and their carers.[1]

The complications of diabetes can be largely prevented or delayed through lifestyle

## INTRODUCTION

Type 2 diabetes mellitus (hereto referred to as diabetes) is among the foremost health challenges facing policy makers in Thailand. Its

change and medication when necessary, and regular screening for early detection and management of complications to control risk factors such as blood glucose, lipids and blood pressure.[4 5] Under Thailand's universal health coverage, nearly everyone diagnosed with diabetes receives timely medical care (>97%) and has access to screening. Yet, surveys suggest that only about half of the people with diabetes achieve optimal control of risk factors or receive annual screening for microvascular complications (53%–60%).[1 6] Limited data support a lack of engagement and self-management skills among those diagnosed with diabetes as the main underlying reasons for this.[7]

Successful management of diabetes involves a considerable degree of self-management. People with diabetes need to adhere to multiple behaviours, including healthy lifestyles, regular monitoring and medication, problem-solving and healthy coping strategies. In this, they are greatly supported by diabetes self-management education (DSME), defined as 'a collaborative and ongoing process intended to facilitate the development of knowledge, skills, and abilities that are required for successful self-management of diabetes'.[8] Evidence from over 100 studies, including many randomised controlled trials conducted predominantly in high-income countries (HIC), suggests that DSME programmes are associated with improvements in a range of behavioural outcomes (knowledge, behaviours, self-efficacy, psychosocial) and clinical outcomes (physiological risk factors, screening for complications, quality of life)[9 10] and are cost-effective.[11] Therefore, DSME programmes are recommended by most clinical guidelines.[8]

However, there is considerable heterogeneity in the effectiveness of DSME programmes.[9 10] Programmes that are more effective usually offer more than 10 hours of contact between trainers and patients, incorporate behavioural approaches and provide longer-term support mechanisms. However, providing intensive and sustained support has cost implications, resulting in ongoing efforts to identify more cost-efficient ways to deliver DSME, notably through use of lay health workers or peer educators, such as Thai village health volunteers (VHV).

Peers can support sustained changes in complex health behaviours by providing assistance in daily management, social and emotional support, linkage to clinical care and ongoing availability of support.[12 13] Unlike the educational/psychological framework of professional support, peer support operates on a social support framework. Although traditionally restricted to those with experience of disease, the definition of peers has been expanded to include other non-professionals with a close relationship with the community (eg, VHV).[14] However, despite widespread interest, empirical data on effectiveness of peers in supporting behaviour change in chronic diseases, including diabetes, are limited and inconsistent.[15 16] In an earlier review, the WHO did not find sufficient evidence to recommend peer support programmes as a policy option for diabetes management in low-income and middle-income countries (LMICs).[17] Whereas many studies on the effectiveness of DSME programmes come from HIC, there is a dearth of data from LMIC settings on cost-effectiveness, acceptability and potential adverse consequences of peer support programmes as well as optimal strategies for mobilising and integrating peers in diabetes care pathways.[13 18 19]

In the Thai healthcare system, structured DSME is not routinely available. While several small-scale studies from Thailand have demonstrated that DSME can strengthen self-management of diabetes, negative perceptions of educational programmes and concerns about the burden on existing staff time and costs have so far prevented the introduction of DSME.[1 19] However, recent policy developments in Thailand are supportive of DSME introduction, if a scalable model can be found. We therefore hypothesise that a nurse-led and/or peer-assisted model for DSME delivery will be effective in improving blood glucose among people with diabetes, with the peer-assisted model being the more scalable option for the Thai healthcare system. We propose to evaluate this through a three-arm cluster randomised controlled trial.

## METHODS AND ANALYSIS
### Study design
This study is an Medical Research Council complex intervention,[20] three-arm cluster randomised controlled trial. Primary care units from within two provinces: Chiang Mai and Lampang will be randomised for patients to receive either the nurse-led or peer-assisted DSME intervention or standard care (brief education session by a nurse). Assessments will be undertaken at baseline, 6-month and 12-month follow-up. A process and cost-effective evaluation will also be conducted.

### Setting and participant selection
Potential participants requiring a DSME intervention will be recruited from 21 primary care units in Chiang Mai (7 primary care units) and Lampang provinces (14 primary care units) in northern Thailand. Chiang Mai is a province of over 1.4 million people with 24 district hospitals and about 250 primary care units. Lampang is a province of approximately 700 000 people with 12 district hospitals and about 140 primary care units. While diabetes is diagnosed at tertiary and district hospitals, it is managed at the primary care unit health centres, which are served by a full-time nurse (doctor visits weekly), and 10–15 VHV linking patients in the community.

From clinical records, we will recruit all new referrals for diabetes management and patients with uncontrolled diabetes diagnosed in the past 3 years at the 21 primary care units over a 9-month period (n=693). Posters and information sheets will be used to provide necessary, trial-related information to prospective participants (figure 1).

Participants presenting to one of the 21 primary care units will be included if they are:

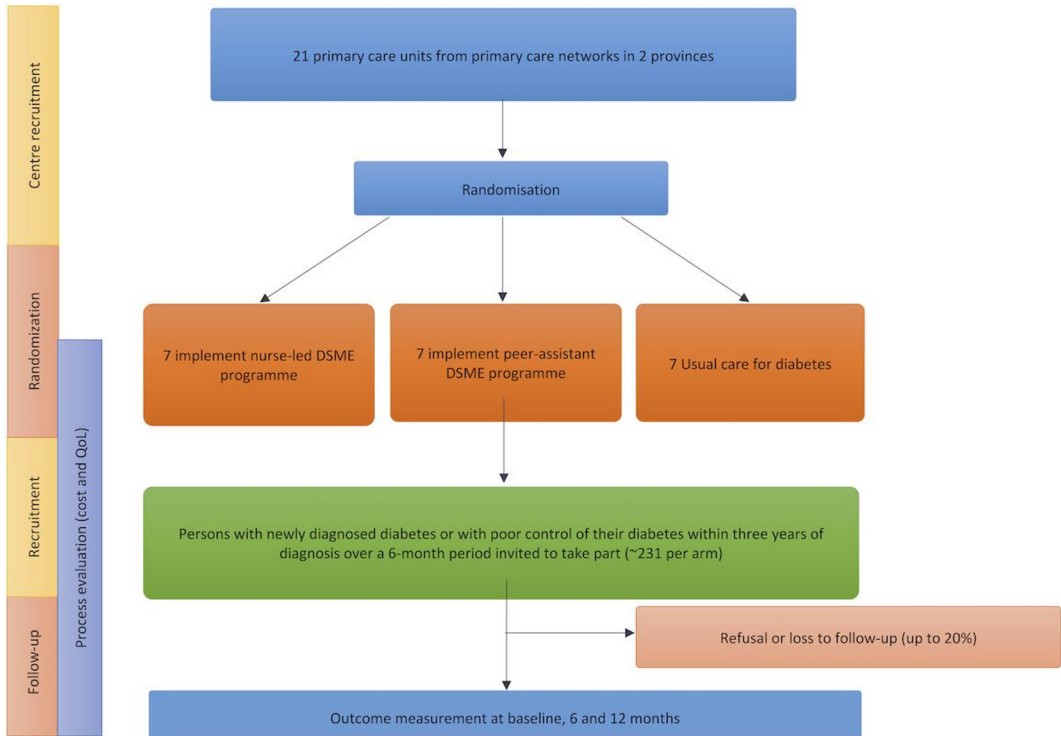

**Figure 1** Trial flow diagram.

1. over 18 years of age with a new referral for type 2 diabetes management;
2. Over 18 years of age with uncontrolled diabetes (HbA1c>7 %) within the first 3 years of diagnosis; willing and able to attend educational group meetings and
3. available for 6-month and 12-month follow-up visits.
   Participants will be excluded if they:
1. have advanced diabetic complications such as diabetic nephropathy, diabetic retinopathy or amputations; or if they are pregnant as patients with these conditions and comorbidities are usually referred to secondary care facilities for treatment in Thailand and not often managed in primary care where this trial is conducted,
2. have learning disabilities, dementia or active severe mental illness or
3. lack the capacity to give voluntary, informed consent.

### Randomisation
Stratified by province, 21 primary care units (7 from Chiang Mai and 14 from Lampang) will be randomised to provide one of three interventions: (1) nurse-led DSME; (2) nurse-led DSME with peer assistance (provided by Thai VHV) or (3) standard care (brief education session by a nurse), resulting in seven primary care units in each arm of the study. All primary care units follow protocols for diabetes management as outlined by national guidelines. Stratification by province will minimise any variation in practice between the different primary care units.

### Sample size calculation
The trial is powered to detect a difference in HbA1c of 0.6% (SD 1.5%) between control and intervention arms, based on the effect size of 0.6% noted in a previous diabetes management study in Thailand,[21] and the fact that an increase in HbA1c of ~0.5% was associated with increased mortality among people with diabetes.[22] An intraclass correlation coefficient between primary care units of 0.02 was assumed based on a similar study which found that the intraclass correlation for HbA1c at 3 years was 0.02 (95% CI 0.00 to 0.08).[23] Allowing for a loss-to-follow up rate of 20%, 693 participants are needed from 21 primary care units (7 in each trial arm arm) to achieve 80% power at 2.5% significance level.

### Informed consent
Written informed consent will be obtained from all study participants in Thai before any study procedures are undertaken including enrolment, intervention allocation, follow-up interviews and blood draws. Local research assistants will explain the study to patients using the patient information sheet (see online supplemental file). The right of the patient to refuse to participate without giving reasons will be respected.

### Intervention description and delivery
The DSME programme has been developed using a structured process. This included a desk and literature review and focus groups with local nurses and VHV to develop a paper prototype of the intervention, including a hypothesised pathway of action (eg, common-sense model of illness, empowerment, discovery learning, social learning and social support) and a training manual for nurses and VHV. In addition, seven brief films (5–6 min long) have been developed and have been used to trigger discussions

on key topic areas during DSME intervention meetings covering such topics as medical adherence, dietary recommendation, physical activity and stress management. Films will be used in the intervention as they are increasingly recognised as a highly effective medium for improved recall when communicating large amounts of information, particularly in low literacy settings.[24] Films will be in the local language and use local people.

Our DSME programme will consist of four modules. Module 1 covers the general overview of diabetes, treatment targets and goal setting. Module 2 covers diet and nutrition. Module 3 covers physical activity and exercise, while module 4 covers stress management and mental health. Each module takes approximately 1.5 hours. Each participant is given an information and self-assessment booklet which covers all content and materials for the four modules.

In addition to routine care, in the nurse-led arm, the nurse will deliver the DSME to groups of 5–10 participants per session within the first months after enrolment. The participants in the nurse-led arm will also be given a refresher session going over all four modules again at 6 months after enrolment. For the peer-assisted arm, a VHV will participate as an assistant to the nurse in the first DSME session. However, the VHV will lead the refresher course at 6 months. In addition, participants in the peer-assisted arm will receive monthly contact with the VHV either via a home visit or telephone call. During these brief 15–20 min monthly contacts, VHV will ask about the progress made, providing encouragement if plans for self-management are being followed or discussing ways to overcome barriers and set new goals if obstacles are identified. Contents of the three trial arms are summarised in table 1.

The intervention will be piloted at four community primary care units with four nurses and four VHV who will be trained to deliver the DSME programme to groups of 5–10 persons with diabetes. This will allow for

**Table 1** Summary of DSME delivery in the three trial arms

| Month | Routine care | Nurse-led DSME | Peer-assisted DSME* |
|---|---|---|---|
| 0 | Individual session | Nurse provides DSME (four modules) | Nurse provides DSME (four modules) with VHV to assist the sessions |
| 6 | Individual session | Refresher course (four modules) provided by nurse | Refresher course (four modules) led by VHV |
| 12 | Outcome assessment | Outcome assessment | Outcome assessment |

*Participants in the peer-assistant arm will additionally receive monthly contact with the VHV either via a home visit or telephone call.
DSME, diabetes self-management education; VHV, village health volunteer.

refining the intervention, ensuring data collection can be completed as specified, and to check our assumptions and processes for the trial. For the main trial, a 2-day workshop will be held such that at least one nurse and one VHV from each primary care unit will be trained by the Thai research team to deliver the DSME programme at the community primary care unit or neighbourhoods as appropriate. The trial coordinator will conduct periodic site visits as additional training as requested and a line of communication will be established between the research team and each site to answer any issues which may arise.

A process evaluation using qualitative methods will be conducted during the trial period and at the end of the study. Observations including video recordings of intervention delivery will be made. We plan to conduct 5–10 focus groups among providers (nurses and VHV) to explore healthcare professionals' perspectives regarding their experience and implementation of the DSME programme, including views on the cultural transferability of DSME and scalability to the Thai context. In addition, we plan to conduct 20 structured interviews with patients. These evaluations will help assess intervention delivery (fidelity, dose and reach), clarify causal mechanisms (those hypothesised by theory of change developed within the project or emergent mechanisms identified) and detail contextual factors (barriers, facilitators) associated with variation in outcomes.[25] The process evaluation will also consist of one-to-one interviews with clinicians and policymakers and direct observations of patients. Data for economic evaluations (resource usage and quality of life using EQ-5D)[26] will be obtained prospectively alongside the trial.

## Standard care

Patients in the control group will receive standard care in the form of a brief didactic educational session at the time of diagnosis of diabetes and during routine clinic visits at 6 months.[27]

## Study outcomes

The two primary outcomes of the intervention are a difference in trial arms at 1-year follow-up in HbA1c and cardiovascular risk score. There is a growing recognition of the importance of combining tight glycaemic control with reduction in other cardiovascular risk factors for prevention of or reduction in complications.[5] The cardiovascular risk will be estimated by the Thai cardiovascular risk score model, which estimates the risk of dying from any cardiovascular disease over 10 years based on age, gender, smoking habits, total cholesterol and systolic blood pressure, as it has been calibrated for use in a Thai population.[28]

Additional secondary outcomes include changes at 1 year for biological, physical, psychosocial, lifestyle and intervention-related measures, as described in table 2.

**Table 2** Summary of primary and secondary outcome measures

| | Measures or questionnaires |
|---|---|
| **Primary outcomes** | |
| Haemoglobin A1c levels (HbA1c) | HbA1c will measure the average blood glucose (sugar) levels over the past 2–3 months |
| Thai Cardiovascular risk score | Estimates the risk of dying from any cardiovascular disease over 10 years based on age, gender, smoking habits, total cholesterol and systolic blood pressure. |
| **Secondary outcomes** | |
| Biological and physical measures | Body weight, body mass index; blood lipids (total, LDL-C HDL, triglycerides), waist circumference, blood pressure, fasting blood glucose |
| Quality of life | WHOQOL-BREF. A 26-item questionnaire developed by WHO to assess quality of life in adults[29] |
| | The European Quality of life questionnaire (EuroQol EQ-5D 5L).[26] EQ-5D is a quality of life measure that includes five quality of life questions on mobility, self-care, usual activity, pain, anxiety/depression and a scale of 0–100 on how the person is feeling on that day. |
| Depression | Hospital Anxiety and Depression Scale (HADS).[30] HADS measures depression and anxiety that will address psychological change with a scale from 0 to 3. |
| Stress | Perceived Stress Questionnaire (PSS).[31] PSS is a psychological instrument for measuring the perception of stress. Ten items with a scale from 1 to 4. |
| Physical activity | International Physical Activity Questionnaire (IPAQ).[32] Short form IPAQ is an assessment of physical activity comprising of seven questions. There are two forms of output from scoring the IPAQ. Results can be reported in categories (low activity levels, moderate activity levels or high activity levels) or as a continuous variable (MET-minutes a week). MET-minutes represent the amount of energy expended carrying out physical activity. |
| Diabetes knowledge and skills | Brief diabetes Illness Perception Questionnaire (B-IPQ).[33] B-IPQ has nine components of which the first five questions assess the cognitive representation of illness perception, two of the questions assess the emotional representation, one item assesses comprehensibility and one item on the root cause of the illness |
| | Diabetes Self-Management Education and Support (DMSES).[34] DMSES is one of the most widely used scale in measuring self-efficacy in type 2 diabetes management. The Thai-DMSES has 20 questions which has been demonstrated to have good psychometric properties[35] |
| | Summary of Diabetes Self-Care Activities questionnaire (SDSCA).[36] SDSCA is a diabetes self-care activities questionnaire focusing on general diet, diabetes-specific diet, physical activity, blood-glucose testing, foot care and smoking. |
| Satisfaction with intervention | Chronic Illness Resources Survey (CIRS).[37] CIRS is a questionnaire to represent patient's received support. Individual's support for behavioural-specific disease management is assessed: proximal support, for example, friend and family and distal factor for example, neighbourhood or community. |
| | Modified Medical Interview Satisfaction Scale (MISS-21).[38] MISS-21 is a questionnaire to measure patient satisfaction with patient and healthcare professional communication/consultation. |

## Data collection and follow-up

Each participant will be involved in the study for 12 months after taking consent and baseline data. The trial is expected to start January 2020 and finish October 2021.

Data collection methods will include:

## Questionnaires

Questionnaire data will be collected face-to-face by research assistants for the full sample at baseline, 6 and 12 months at the community primary care unit where participants are recruited. A custom-designed form linked to RedCap will be used to collect, validate, verify and store respondents' data where possible or else data will be collected via paper forms and double-entered into the databases. All data files and databases will be password protected.

## Biological samples

Blood samples will be collected at baseline, 6 and 12 months, to measure fasting blood glucose, HbA1c and lipids, coordinating where possible with the annual routine tests offered to patients to reduce duplication. All blood samples will be administered from participants by trained phlebotomists. Data will be linked to the participant information using a unique respondent ID, which will be assigned to all study participants.

## Interviews

During the delivery of the intervention, a process evaluation using a subset of participants will be conducted using in-depth interviews and focus group discussions. These will be audio-recorded. Data will be collected using a range of qualitative methods: (a) one-to-one interviews and focus

group discussions with nurses, health volunteers, people with diabetes and their carers (5–10 focus groups and 20 semistructured interviews) and (b) ethnography through direct observations including video recordings of intervention delivery and unstructured interviews with clinical managers and policy makers. The data collected will be used to capture the range of experiences of the intervention and identify unanticipated pathways to generate new theories as well as exploring the scalability of the intervention.

### Trial follow-up appointments

The research team will hold weekly briefings with the study coordinators to generate a list of priority areas and loss to follow-up participant lists. Arrangements to follow-up participants who have not turned up for their appointment will be made, with attempts to contact participants through SMS, phone calls or house visits. Participants will be declared lost to follow-up if they do not show for a month and are untraceable.

### Data management

A data collection protocol will be developed, and the study coordinator in Thailand will provide training to fieldworkers before data collection commences. Validation will be performed on a random sample of questionnaire data by crosschecking with clinic records. Any discrepancies will be followed-up and addressed by field workers, recontacting participants to clarify as necessary. Using Redcap (https://redcap.med.cmu.ac.th), quantitative data will be entered directly via a form with built-in data checks to minimise transcription errors (or where necessary collected on paper and later double entered into the electronic form). Postentry checks will be conducted by exploring the distribution, ranges and outliers of each variable. All hospital laboratories have their own internal quality assurance protocols and are also linked to a national external quality assurance mechanism. Fieldworkers will be trained in qualitative methods and an interview schedule will be devised.

### Statistical analysis

#### Quantitative analysis

Available outcome data will be analysed on an intention-to-treat basis. Potential clustering of outcomes (HbA1c at 12 months and cardiovascular disease risk score at 12 months) at the level of community primary care units will be accounted for using random intercept models. To improve precision of the estimates, outcomes will be adjusted for their baseline values. In case of baseline imbalances of relevant covariates (eg, age, education level, body mass index), judged by statistical significance at $p<0.05$, we will conduct a secondary analysis adjusting for these covariates.

#### Qualitative analysis

Qualitative data from interviews, focus groups and direct observations will be transcribed and analysed using NVivo software. The data will be analysed using a descriptive, phenomenological approach to understand participants' experiences and interpret them within their respective cultural contexts. Comparative analysis will compare and contrast these themes across participants. Deviant cases will be actively sought throughout the analysis and emerging ideas and themes modified in response. In addition, thematic analysis will be used to inform elements of scalability and to produce a set of considerations in making decisions about the scalability of the intervention.

### Cost-effective analysis

Data for economic evaluation (resource usage and quality of life using EQ-5D) will be obtained prospectively alongside the trial. We will aim to capture all health service contacts, as well as out-of-pocket expenses and medication use. Educator training costs will be included, as well as minimal intervention material costs (as most will be made available freely after the trial). Utility values from EQ-5D will be derived using a Thai tariff. Incremental cost utility will be estimated from the Thai health system and societal perspectives to provide incremental cost-effectiveness ratio and probability of being cost-effective at Thai government's willingness to pay threshold of 160 000 baht/QALY.

## ETHICS AND DISSEMINATION

This study is to be conducted according to the international standards of Good Clinical Practice (International Conference on Harmonization guidelines), Declaration of Helsinki and International Ethical Guidelines for Biomedical Research Involving Human Subjects, applicable national government regulations and institutional research policies and procedures. All investigators received Good Clinical Practice training at the onset of the study. Ethical approval was obtained prior to start of the project from Chiang Mai University (No. 326/2018) and the London School of Hygiene & Tropical Medicine (16113/RR/12850). The study protocol, informed consent form, patient information sheet and other relevant information has been approved. Any future amendments of the protocol shall be submitted to and approved by the Institutional Review Board (IRB) before implementation.

### Trial monitoring and oversight

The role of the Trial Steering Committee (TSC) is to provide overall supervision of the trial. The TSC will meet every 6 months. The TSC will include experts in the field of DSME, health psychology and clinical trials as well as an independent Chair.

### Dissemination

Research findings will be disseminated to scientific audiences at major conferences and published in high-impact, open-access scientific journals; planned publications include those on intervention development, primary trial

results, process evaluation and health systems analysis, at a minimum. This study is expected to have a major policy impact due to the close involvement of a key policy maker in the project. Towards the end of the study, a dedicated workshop will be held with key governmental stakeholders to disseminate the recommended model for DSME implementation in Thailand and encourage inclusion of a large-scale scientific evaluation into any national implementation of the scheme.

**Author affiliations**
[1]Department of Family Medicine, Faculty of Medicine, Chiang Mai University, Chiang Mai, Thailand
[2]Department of Non-Communicable Disease Epidemiology, Faculty of Epidemiology and Population Health,London School of Hygiene and Tropical Medicine, London, UK
[3]ASEAN Institute for Health Development, Mahidol University, Salaya, Thailand
[4]Department of Surgery, Faculty of Medicine, Chiang Mai University, Chiang Mai, Thailand
[5]Research Institute for Health Sciences, Chiang Mai University, Chiang Mai, Thailand
[6]Royal Thai Government Ministry of Public Health, Bangkok, Thailand
[7]Lampang Hospital, Lampang, Thailand
[8]Surat Thani Rajabhat University, Surat Thani, Thailand
[9]Department of Global Health and Development, Faculty of Public Health of Public Health and Policy, London School of Hygiene and Tropical Medicine, London, UK
[10]Department of Health Sciences, University of Leicester, Leicester, UK

**Contributors** CA, SS, OQ, PACM and SK were involved in conception and trial design. CA, KP, PACM and SK wrote the first draft of the trial design proposal. KR, WT, KH and KK were involved in critical revision of the trial design proposal. All authors contributed to finalising the trial protocol. CA, KP, IPN, CP, AHH and NW were involved with preparation of the manuscript for submission. All authors reviewed the manuscript and approved the manuscript for publication.

**Funding** This study is supported by UK Medical Research Council (MRC) grant number (MR/R020876/1) and the Thailand Research Fund (TRF) grant number (DBG6180007).

**Competing interests** None declared.

**Patient and public involvement** Patients and/or the public were not involved in the design, or conduct, or reporting, or dissemination plans of this research.

**Patient consent for publication** Not required.

**Provenance and peer review** Not commissioned; externally peer reviewed.

**ORCID iD**
Chaisiri Angkurawaranon http://orcid.org/0000-0003-4206-9164

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
