## [Reviewer comments · BMJ Open]

ARTICLE DETAILS

TITLE (PROVISIONAL)	A Scalable Solution for Delivery of Diabetes Self-Management Education in Thailand (DSME-T): A Cluster Randomized Trial Study Protocol
AUTHORS	Angkurawaranon, Chaisiri; Papachristou Nadal, Iliatha; Mallinson, Poppy Alice Carson; Pinyopornpanish, Kanokporn; Quansri, Orawan; Rerkasem, Kittipan; Srivanichakorn, Supattra; Techakehakij, Win; Wichit, Nutchanath; Pateekhum, Chanapat; Hashmi, Ahmar H.; Hanson, Kara; Khunti, Kamlesh; Kinra, Sanjay

VERSION 1 – REVIEW

REVIEWER	Arlene Smaldone Columbia University, United States
REVIEW RETURNED	01-Feb-2020

GENERAL COMMENTS	Thank you for the opportunity to review this study protocol. The purpose of this study is to test a culturally tailored DSME intervention using a 3-arm clustered randomized controlled trial that compares a nurse-led intervention, a peer assisted model, and standard care for both effectiveness of the intervention at 12 months and cost effectiveness. Outcome measures are HbA1c and cardiovascular risk. The topic is important as the prevalence of type 2 diabetes has more than tripled over the past 20 years. However, in its current format, sufficient detail regarding the intervention and the analytic strategies is lacking. My specific comments follow. Overall comment - the manuscript should be checked for misspellings, for example, DMSE (page 5) enrollment (page 11), and grammatical errors Abstract: Ethics and dissemination - states that ethical approval was submitted - yet later in the manuscript, wording implies that approval has been obtained Methods and Analysis Study design - need to include the control arm within the description Randomization - method of randomization needs to be stated. The authors need to be clearer about what they mean by the last sentence in the randomization paragraph. Sample size calculation - intra-cluster correlation coefficient between primary care units needs a reference to support the coefficient selected. Intervention description and delivery
--

	Please note within the manuscript where Figure 1 should be included. The intervention needs to be more clearly described in terms of how many sessions over what period of time, topics covered within each session, how the intervention is culturally tailored and the size of intended group. Will people who have lived with diabetes longer be included in groups of newly referred adults? A table outlining the intervention and any differences, if any, between the nurse led and peer assisted models. How will those who deliver the intervention be trained. How will consistency in delivery of the intervention be managed? Statistical analysis: Quantitative analysis - the statistical methods for quantitative analysis need to be more fully described. For example, is HbA1c at 12 months or the change in HbA1c from baseline to 12 months the outcome of interest. The authors also need to be clearer regarding how baseline imbalances in outcomes will be determined. A section regarding the approach for the cost effectiveness analysis needs to be added. Currently, it is missing from the manuscript. Data management - The authors need to be clearer regarding what they mean by how post entry checks will be conducted using statistical software.
--	---

REVIEWER	Dr Elizabeth Morris University of Oxford, UK
REVIEW RETURNED	01-Mar-2020

GENERAL COMMENTS	Thank you for inviting me to review this interesting and topical study protocol. The authors are hoping to conduct a study to gather robust and clinically relevant data to inform local and national guidance in a relatively under-researched patient population. I am looking forward to reading the results of this study once it is complete. I have several comments, the majority of which request expansion on the details given in this manuscript, which I hope will add clarity and value for the readers of this paper. Major comments: Randomisation: Please give details of how randomisation will be performed. You state 21 primary care units in 2 provinces will be included- is this all the units in this region, or a selection (and if so, what proportion of the total number of primary care units is this)? How will the participating primary care units be recruited – have they already been identified? Will there be any stratification in the randomisation, e.g. on practice size, or demographics? Sample size calculation: Please describe or reference how you defined 0.6 units as clinically important (and which “units” of HbA1c are being used – DCCT units (%) / IFCC units (mmol/mol), or other (mg/dl is referenced later on?)); Given the threshold of 0.6 I would presume you mean 0.6%? Please clarify how you calculated - or on what basis assumed - the ICC to be 0.02 – the references you cite are 1) the DESMOND study
---

– which assumed an ICC of 0.05, and 2) an interesting study of DSME in a Thai population but that does not seem to have been cluster randomised.

Intervention development:

One of the main indications to publish a protocol paper of a novel complex intervention such as this, can be to share details and methods of the intervention development, for which there may not be space in the main paper. In this section (line 155) you say the intervention was developed using a “structured process”. Were there any specific or previously described intervention development processes/methods you used? If so, please describe and cite. Either way, please give more information about the intervention development: What principles for the intervention/intervention design were established from the “desk and literature review”? What was the starting point for the focus groups (was this an engagement exercise, reviewing a prototype already developed by the research team; or was it generating the key components that would then be included in the final intervention documents?). How were the presumed theories of education by which it could be facilitated identified (e.g. discovery learning theory, social learning theory)? Were there any behaviour change theories or strategies incorporated, to target healthcare professional, VHV, or patient behaviours?

Line 168- You describe that “the intervention will be piloted.. refining the intervention, ensuring data collection can be completed as specified, and to check our assumptions and processes for the trial”. Is this an internal pilot study, in which case are there pre-specified outcomes to guide whether the full scale trial progresses, or that would indicate a substantial change in the protocol or methods is needed? If so please give details. You have listed the anticipated start date of the trial as Jan 2020 – in which case has the pilot already been conducted, and will the outcomes be published or incorporated into this study protocol?

Line 174 –You say you will assess intervention fidelity as part of a process evaluation. What are the outcome measures through which you will assess fidelity? Please include any prespecified fidelity outcome measures here. You describe that this will be conducted via interviews with clinicians and observation of patients - do you think these methods of data collection will adequately capture fidelity of intervention delivery? You mention later (line 216 onwards) videoing sessions for qualitative data collection – could this also be used to extract relevant quantitative data regarding whether fidelity criteria for intervention delivery are met (e.g. has specific content been included, whether factual/medical information, or behaviour change techniques)? You also say that “dose” of intervention delivery will be assessed as part of the process evaluation – will you be using an existing framework to report this?

Study outcomes (line 185):

You state what outcomes will be measured but other than for HbA1c do not specify how they will be assessed and reported (please state if this is the same for all outcomes as for HbA1c- difference between trial arms at one-year follow-up?).

It is not clear from your table whether “cardiovascular risk assessment” via the Thai CV risk score model is your sole secondary outcome in the “CV risk” target domain, or whether each of the measures listed (HbA1c, lipids, BMI, waist circumference, blood pressure) are also additional secondary outcomes. Did you consider reporting weight change, as well as BMI change, as a

	secondary outcome? If you are already measuring waist circumference, did you consider also doing hip circumference to enable calculation of waist:hip ratio (WHO concluded some utility in both measures over and above BMI for assessing body fat distribution and NCD risk)? You state in your abstract, strengths and study design sections that a cost-effectiveness evaluation will be completed as part of the study, but this is not then mentioned in the study outcomes or the statistical analysis sections. Please expand on what outcome data will be collected for this modelling and provide any relevant details of the analysis plan. Minor comments: Abstract, line 6 – typo – abbreviation should read DSME (not DMSE). Line 46 – I would suggest changing “diabetic people” to “people with diabetes” here, to fit with the otherwise predominant use of person-first language throughout, which was noted and appreciated. Line 134 – Definition of uncontrolled diabetes – are these units correct? Do you mean 7 mg/dl (as written) - or maybe 7%? Line 196, Table 1 is entitled “primary and secondary outcome measures”; please make a distinction in the table between which are primary outcomes and secondary outcome measures. Ethical approval: You describe 3 different states of ethics approval throughout the manuscript; abstract line 20 “ethical approval was submitted”, line 267 “ethical approvals will be sought”, line 314 “the study has been approved”. I suspect this may just be due to multiple drafts throughout the preparation of the protocol but please update the wording earlier on (if ethical approval has now been granted) in order to be consistent throughout.
--	--

VERSION 1 – AUTHOR RESPONSE

Reviewer(s)' Comments to Author:

Reviewer: 1

Reviewer Name: Arlene Smaldone

Institution and Country: Columbia University, United States

Please state any competing interests or state 'None declared': None declared

Please leave your comments for the authors below

Thank you for the opportunity to review this study protocol. The purpose of this study is to test a culturally tailored DSME intervention using a 3-arm clustered randomized controlled trial that compares a nurse-led intervention, a peer assisted model, and standard care for both effectiveness of the intervention at 12 months and cost effectiveness. Outcome measures are HbA1c and cardiovascular risk.

The topic is important as the prevalence of type 2 diabetes has more than tripled over the past 20 years. However, in its current format, sufficient detail regarding the intervention and the analytic strategies is lacking. My specific comments follow.

Comment 1: Overall comment: the manuscript should be checked for misspellings, for example, DMSE (page 5) enrolment (page 11), and grammatical errors

We have double checked our manuscript for misspellings and grammatical errors.

Comment 2) Abstract: Ethics and dissemination- states that ethical approval was submitted - yet later in the manuscript, wording implies that approval has been obtained

Ethical approval was obtained prior to commencement of the project from Chiang Mai University [No 326/2018] and the London School of Hygiene & Tropical Medicine [16113/RR/12850]. We have updated our abstract and have added the approval reference number to the main text. (Lines 315-317)

Comment 3) Methods and Analysis- Study design need to include the control arm within the description

We have updated our text "This study is a MRC complex intervention,¹⁹ three-arm cluster randomised controlled trial. Primary care units from within two provinces: Chiang Mai and Lampang will be randomised for patients to receive either the nurse-led or peer-assisted DSME intervention or standard care (brief education session by a nurse). Assessments will be undertaken at baseline, 6- and 12-month follow up. A process and cost-effective evaluation will also be conducted." (lines 99-103)

Comment 4) Randomization - method of randomization needs to be stated. The authors need to be clearer about what they mean by the last sentence in the randomization paragraph.

We have clarified that we stratified by province as there were potential variations in routine diabetes care between province.

The text now reads "Stratified by province, 21 primary care units (7 from Chiang Mai and 14 from Lampang) will be randomised to provide one of three interventions: (1) nurse-led DSME; (2) nurse-led DSME with peer assistance (provided by Thai village health volunteers, VHV); or (3) standard care (brief education session by a nurse), resulting in seven primary care units in each arm of the study. All primary care units follow protocols for diabetes management as outlined by national guidelines. Randomisation by province will minimise any variation in practice between the different primary care units." (lines 131-138)

Comment 5) Sample size calculation intra-cluster correlation coefficient between primary care units needs a reference to support the coefficient selected.

We have provided a reference and clarified that the ICC of 0.02 was based on the observed value of the a previous report that the intraclass correlation for HbA1c at three years was 0.02 (95% confidence interval 0.00 to 0.08)

We state “An ICC between hospitals of 0.02 was assumed based on a similar study which found that The intraclass correlation for HbA1c at three years was 0.02 (95% confidence interval 0.00 to 0.08²².”(lines 144-146)

Comment 6) Intervention description and delivery- Please note within the manuscript where Figure 1 should be included.

We have now added a note to refer to figure 1 in the main text (line 117)

Comment 7) The intervention needs to be more clearly described in terms of how many sessions over what period of time, topics covered within each session, how the intervention is culturally tailored and the size of intended group. Will people who have lived with diabetes longer be included in groups of newly referred adults? A table outlining the intervention and any differences, if any, between the nurse led and peer assisted models.

We are planning to write a detailed paper on how our intervention has been developed and culturally tailored. For this protocol paper, we have now added a description of the intervention in the main text and as a table1

“Our DSME programme will consist of 4 modules. Module 1 covers the general overview of diabetes, treatment targets and goal settings. Module 2 covers diet and nutrition. Module 3 covers physical activity and exercise while module 4 covers stress management and mental health. Each module takes approximately 1.5 hours. Each participant is given an information and self-assessment booklet which covers all contents and materials for the four modules.

In addition to routine care, in the nurse-led arm, the nurse will deliver the DSME to groups of 5-10 participants per session within the first months after enrolment. The participants in the nurse-led arm will also be given a refresher session going over all 4 modules again at 6 months after enrolment. For the peer-assisted arm, a peer volunteer will participate as an assistant to the nurse in the first DSME session. However, the peer will lead the refresher course at 6 months. In addition, participants in the peer-led arm will received monthly contact with the peer either via a home visit or telephone call. During these brief 15-20 minute monthly contacts, peers will ask about the progress made, providing encouragement if plans for self-management are being followed or discussing ways to overcome barriers and set new goals if obstacles are identified. (Table 1)” (line 170-184)

Table 1 Summary DSME Delivery

Month	Routine care	Nurse-led DSME	Peer-assisted DSME**
0	Individual session	Nurse provides DSME (4 modules)	Nurse provides DSME (4 modules) with peer volunteers to assist the sessions
6	Individual session	Refresher course (4 modules) provided by	Refresher course (4 modules) led by peers

		nurse	
12	Outcome assessment	Outcome assessment	Outcome assessment

* participants in the peer-led arm will received monthly contact with the peer either via a home visit or telephone call.

Comment 8) How will those who deliver the intervention be trained. How will consistency in delivery of the intervention be managed?

Thank you for your comment. We have clarified in the text

“For the main trial, a two-day workshop will be held such that at least one nurse and one VHV from each primary care unit will be trained by the Thai research team to deliver the DSME programme at the community hospital or neighbourhoods as appropriate. The trial coordinator will conduct periodic site visits as additional training as requested and a line of communication will be established between the research team and each site to answer any issues which may arise.” (line 192-197)

A process evaluation using qualitative methods will be conducted during the trial period and at the end of the study. Observations including video recordings of intervention delivery will be made. We plan to conduct 5-10 focus group among providers (nurses and village health volunteers) to explore healthcare professionals’ perspectives regarding their experience and implementation of the DSME programme, including views on the cultural transferability of DSME and scalability to the Thai context. In addition, we planned to conduct 20 structured interviews with patients. These evaluations will help assess intervention delivery (fidelity, dose and reach), clarify causal mechanisms (those hypothesised by theory of change developed within the project or emergent mechanisms identified), and detail contextual factors (barriers, facilitators) associated with variation in outcomes.²⁴” (line 199-208)

Comment 9) Statistical analysis- Quantitative analysis - the statistical methods for quantitative analysis need to be more fully described. For example, is HbA1c at 12 months or the change in HbA1c from baseline to 12 months the outcome of interest. The authors also need to be clearer regarding how baseline imbalances in outcomes will be determined.

We have clarified “Potential clustering of outcomes (HbA1c at 12 months and CVD risk score at 12 months) at the level of community primary care units will be accounted for using random intercept models. and adjusted for baseline values. To improve precisions of the estimates, outcomes will be adjusted for their baseline values. In case of baseline imbalances of relevant covariates (e.g. age, education level, body mass index), judged by statistical significance at $p < 0.05$, we will conduct a secondary analysis adjusting for these covariates.” (lines 279-285)

Comment 10) A section regarding the approach for the cost effectiveness analysis needs to be added. Currently, it is missing from the manuscript.

Response We have added “Data for economic evaluation (resource usage and quality of life using EQ-5D) will be obtained prospectively alongside the trial. We will aim to capture all health service contacts, as well as out-of-pocket expenses and medication use. Educator training costs will be included, as well as minimal intervention material costs (as most will be made available freely after the trial). Utility values from EQ-5D will be derived using a Thai tariff. Incremental cost utility will be estimated from the Thai health system and societal perspectives, to provide incremental cost-effectiveness ratio and probability of being cost-effective at Thai government’s willingness to pay threshold of 160,000 baht/QALY.” (lines 297-305)

Comment 11) Data management - The authors need to be clearer regarding what they mean by how post entry checks will be conducted using statistical software.

Thank you, we have now further clarified this “Using Redcap (<https://redcap.med.cmu.ac.th>), quantitative data will be entered directly via a form with built-in data checks to minimise transcription errors (or where necessary collected on paper and later double entered into the electronic form). Post entry checks will be conducted by exploring the distribution, ranges, and outliers of each variable.” (lines 269-2

Reviewer: 2

Reviewer Name: Dr Elizabeth Morris

Institution and Country: University of Oxford, UK

Please state any competing interests or state ‘None declared’: None declared

Please leave your comments for the authors below

Thank you for inviting me to review this interesting and topical study protocol. The authors are hoping to conduct a study to gather robust and clinically relevant data to inform local and national guidance in a relatively under-researched patient population. I am looking forward to reading the results of this study once it is complete.

I have several comments, the majority of which request expansion on the details given in this manuscript, which I hope will add clarity and value for the readers of this paper.

Major comments:

Comment 1) Randomisation: Please give details of how randomisation will be performed. You state 21 primary care units in 2 provinces will be included- is this all the units in this region, or a selection (and if so, what proportion of the total number of primary care units is this)? How will the participating primary care units be recruited – have they already been identified? Will there be any stratification in the randomisation, e.g. on practice size, or demographics?

Please see our response to reviewer 1 (comment 4) about the issue of randomization. We stratified by province to reduce variations in routine practices between province. In addition we have clarified in the text

“Potential participants requiring a DSME intervention will be recruited from 21 primary care units in Chiang Mai (7 primary care units) and Lampang provinces (14 primary care units) in northern Thailand. Chiang Mai is a province of over 1.4 million people with 24 district hospitals and about 250 primary care units. Lampang is a province of approximately 700,000 people with 12 district hospitals and about 140 primary care units. While diabetes is diagnosed at tertiary and district hospitals, it is managed at the primary care unit health centres, which are served by a full-time nurse (doctor visits weekly), and 10-15 village health volunteers (VHV) linking patients in the community.” (lines 105-113)

From clinical records, we will recruit all new referrals for diabetes management and patients with uncontrolled diabetes diagnosed in the past 3 years at the 21 primary care units over a 9-month period (N=693). Posters and information sheets will be used to provide necessary, trial-related information to prospective participants (Figure 1)” (lines 114-117)

Comment 2 Sample size calculation: Please describe or reference how you defined 0.6 units as clinically important (and which “units” of HbA1c are being used – DCCT units (%) / IFCC units (mmol/mol), or other (mg/dl is referenced later on?)); Given the threshold of 0.6 I would presume you mean 0.6%? Please clarify how you calculated - or on what basis assumed - the ICC to be 0.02 – the references you cite are 1) the DESMOND study – which assumed an ICC of 0.05, and 2) an interesting study of DSME in a Thai population but that does not seem to have been cluster randomised.

Thank you for your comment and we have clarified in text

“The trial is powered to detect a difference in HbA1c of 0.6% (SD 1.5%) between control and intervention arms, based on the effect size of 0.6% noted in a previous diabetes management study in Thailand²⁰, and the fact that an increase in HbA1c of ~0.5% was associated with increased mortality among people with diabetes²¹. An ICC between hospitals of 0.02 was assumed based on a similar study which found that The intraclass correlation for HbA1c at three years was 0.02 (95% confidence interval 0.00 to 0.08)²². Allowing for a loss to follow up rate of 20%, 693 participants are needed from 21 primary care units (7 in each trial arm arm) to achieve 80% power at 2.5% significance level.” (lines 140-148)

Comment 3) Intervention development:

One of the main indications to publish a protocol paper of a novel complex intervention such as this, can be to share details and methods of the intervention development, for which there may not be space in the main paper. In this section (line 155) you say the intervention was developed using a “structured process”. Were there any specific or previously described intervention development processes/methods you used? If so, please describe and cite.

Either way, please give more information about the intervention development: What principles for the intervention/intervention design were established from the “desk and literature review”? What was the starting point for the focus groups (was this an engagement exercise, reviewing a prototype already developed by the research team; or was it generating the key components that would then be included in the final intervention documents?). How were the presumed theories of education by which it could be facilitated identified (e.g. discovery learning theory, social learning theory)? Were there any behaviour change theories or strategies incorporated, to target healthcare professional, VHV, or patient behaviours? Line 168- You describe that “the intervention will be piloted.. refining the intervention, ensuring data collection can be completed as specified, and to check our assumptions and processes for the trial”. Is this an internal pilot study, in which case are there pre-specified outcomes to guide whether the full scale trial progresses, or that would indicate a substantial change in the protocol or methods is needed? If so please give details.

Thank you for your comment and you have rightly identified that there is not enough space for the protocol paper to detail the development aspect of the intervention and we are planning to publish a separate paper regarding the development of the intervention. However, as also requested by

reviewer 1, we have described the intervention in the text and a summary of the interventions is presented as table 1 (please see response to comment 7 by reviewer 1)

Comment 4) You have listed the anticipated start date of the trial as Jan 2020 – in which case has the pilot already been conducted, and will the outcomes be published or incorporated into this study protocol?

Yes, we have conducted the pilot and the results of the pilot leading up to the DSME program described will be written as a separate paper as stated in the previous response.

Comment 5) Line 174 –You say you will assess intervention fidelity as part of a process evaluation. What are the outcome measures through which you will assess fidelity? Please include any prespecified fidelity outcome measures here. You describe that this will be conducted via interviews with clinicians and observation of patients - do you think these methods of data collection will adequately capture fidelity of intervention delivery? You mention later (line 216 onwards) videoing sessions for qualitative data collection – could this also be used to extract relevant quantitative data regarding whether fidelity criteria for intervention delivery are met (e.g. has specific content been included, whether factual/medical information, or behaviour change techniques)? You also say that “dose” of intervention delivery will be assessed as part of the process evaluation – will you be using an existing framework to report this?

We have now added information that fidelity assessments and methods used.

“A process evaluation using qualitative methods will be conducted during the trial period and at the end of the study. Observations including video recordings of intervention delivery will be made. We plan to conduct 5-10 focus group among providers (nurses and village health volunteers) to explore healthcare professionals’ perspectives regarding their experience and implementation of the DSME programme, including views on the cultural transferability of DSME and scalability to the Thai context. In addition, we planned to conduct 20 structured interviews with patients. These evaluations will help assess intervention delivery (fidelity, dose and reach), clarify causal mechanisms (those hypothesised by theory of change developed within the project or emergent mechanisms identified), and detail contextual factors (barriers, facilitators) associated with variation in outcomes.²⁴ The process evaluation will also consist of one-to-one interviews with clinicians and policy makers and direct observations of patients. Data for economic evaluations (resource usage and quality of life using EQ-5D²⁵) will be obtained prospectively alongside the trial.” (lines 199-211)

Comment 6) Study outcomes (line 185): You state what outcomes will be measured but other than for HbA1c do not specify how they will be assessed and reported (please state if this is the same for all outcomes as for HbA1c- difference between trial arms at one-year follow-up?).

We have clarified that “Additional secondary outcomes include changes at one year for biophysical data, psychosocial and lifestyle data and intervention related data, as described in Table 2.” (lines 224-226)

Comment 7) It is not clear from your table whether “cardiovascular risk assessment” via the Thai CV risk score model is your sole secondary outcome in the “CV risk” target domain, or whether each of the measures listed (HbA1c, lipids, BMI, waist circumference, blood pressure) are also additional secondary outcomes.

We have clarified that we have two primary outcomes: Hba1C and CV risk score (lines 218-219)

Comment 8) Did you consider reporting weight change, as well as BMI change, as a secondary outcome? If you are already measuring waist circumference, did you consider also doing hip circumference to enable calculation of waist:hip ratio (WHO concluded some utility in both measures over and above BMI for assessing body fat distribution and NCD risk)?

Thank you, as we have already considered BMI change, we will also consider weight change as suggested (table 2). However, as the trial is already underway and we did not measure hip circumference, it is not possible to add waist:hip ratio as part of our assessment.

Comment 9) You state in your abstract, strengths and study design sections that a cost-effectiveness evaluation will be completed as part of the study, but this is not then mentioned in the study outcomes or the statistical analysis sections. Please expand on what outcome data will be collected for this modelling and provide any relevant details of the analysis plan.

This section has been added stating “Data for economic evaluation (resource usage and quality of life using EQ-5D) will be obtained prospectively alongside the trial. We will aim to capture all health service contacts, as well as out-of-pocket expenses and medication use. Educator training costs will be included, as well as minimal intervention material costs (as most will be made available freely after the trial). Utility values from EQ-5D will be derived using a Thai tariff. Incremental cost utility will be estimated from the Thai health system and societal perspectives, to provide incremental cost-effectiveness ratio and probability of being cost-effective at Thai government’s willingness to pay threshold of 160,000 baht/QALY.”(lines 297-305)

Minor comments:

Abstract, line 6 – typo – abbreviation should read DSME (not DMSE).

Corrected

Line 46 – I would suggest changing “diabetic people” to “people with diabetes” here, to fit with the otherwise predominant use of person-first language throughout, which was noted and appreciated.

Corrected (lines 52-53)

Line 134 – Definition of uncontrolled diabetes – are these units correct? Do you mean 7 mg/dl (as written) - or maybe 7%?

Correct to 7% (line 141)

Line 196, Table 1 is entitled “primary and secondary outcome measures”; please make a distinction in the table between which are primary outcomes and secondary outcome measures.

Has been corrected as suggested. (It is now table 2)

Ethical approval: You describe 3 different states of ethics approval throughout the manuscript; abstract line 20 “ethical approval was submitted”, line 267 “ethical approvals will be sought”, line 314 “the study has been approved”. I suspect this may just be due to multiple drafts throughout the preparation of the protocol but please update the wording earlier on (if ethical approval has now been granted) in order to be consistent throughout.

Ethical approval has been granted. The approval number has been added and edits are made so it is consistent throughout

VERSION 2 – REVIEW

REVIEWER	Elizabeth Morris University of Oxford, UK
REVIEW RETURNED	16-Jul-2020

GENERAL COMMENTS	Thank you for the opportunity to re-review this interesting study protocol. I note the authors' responses and the additions to the manuscript which have strengthened its content, in particular detailing the structure of the intervention, and support its publication. I have a few brief general comments: A further proof-read of the document for spelling, grammar, and readability might be helpful (capitalisation “The” mid-line 145, repetition of half a sentence line 281 “adjusted for baseline values”) Exclusions (p7) – the authors may wish to justify why any participant with diabetic complications such as nephropathy, retinopathy, or amputation were excluded from this structured self-management intervention when presumably this would not preclude their participation in education and self-management in routine care, and these patients would often benefit from additional support. Randomisation (line 137) – the authors have stated “randomisation by province” where “stratification by province” might be more consistent with the clarified wording in the paragraph above, if this is what is meant.
--

VERSION 2 – AUTHOR RESPONSE

Reviewer(s)' Comments to Author:

Reviewer: 2

Comment 1) Thank you for the opportunity to re-review this interesting study protocol. I note the authors' responses and the additions to the manuscript which have strengthened its content, in particular detailing the structure of the intervention, and support its publication.

Thank you for your supportive comments

Comment 2) A further proof-read of the document for spelling, grammar, and readability might be helpful (capitalisation "The" mid-line 145, repetition of half a sentence line 281 "adjusted for baseline values")

We have double checked our manuscript for misspellings and grammatical errors.

Comment 3) Exclusions (p7) – the authors may wish to justify why any participant with diabetic complications such as nephropathy, retinopathy, or amputation were excluded from this structured self-management intervention when presumably this would not preclude their participation in education and self-management in routine care, and these patients would often benefit from additional support.

Thank you for your comment. We have clarified that our reason for exclusion since "patients with these conditions and co-morbidities are usually referred to secondary care facilities for treatment in Thailand and not managed in primary care where this trial is conducted" (page 7 lines 128-130)

Comment 4) Randomisation (line 137) – the authors have stated "randomisation by province" where "stratification by province" might be more consistent with the clarified wording in the paragraph above, if this is what is meant.

Thank you for your suggestion. We have edited the text as suggested. (page 8 line 140)